# Impact of Chronic Kidney Disease on Corneal Neuroimmune Features in Type 2 Diabetes

**DOI:** 10.3390/jcm12010016

**Published:** 2022-12-20

**Authors:** Kofi Asiedu, Maria Markoulli, Shyam Sunder Tummanapalli, Jeremy Chung Bo Chiang, Sultan Alotaibi, Leiao Leon Wang, Roshan Dhanapalaratnam, Natalie Kwai, Ann Poynten, Arun V. Krishnan

**Affiliations:** 1School of Optometry & Vision Science, University of New South Wales, Sydney, NSW 2052, Australia; 2School of Clinical Medicine, University of New South Wales, Sydney, NSW 2052, Australia; 3School of Medical Sciences, University of Sydney, Sydney, NSW 2006, Australia; 4Department of Endocrinology, Prince of Wales Hospital, Sydney, NSW 2031, Australia

**Keywords:** diabetes, chronic kidney disease, corneal nerves, dendritic cells, neuroinflammation

## Abstract

Aim: To determine the impact of chronic kidney disease on corneal nerve measures and dendritic cell counts in type 2 diabetes. Methods: In vivo corneal confocal microscopy images were used to estimate corneal nerve parameters and compared in people with type 2 diabetes with chronic kidney disease (T2DM-CKD) (n = 29) and those with type 2 diabetes without chronic kidney disease (T2DM-no CKD) (n = 29), along with 30 healthy controls. Corneal dendritic cell densities were compared between people with T2DM-CKD and those with T2DM-no CKD. The groups were matched for neuropathy status. Results: There was a significant difference in corneal nerve fiber density (*p* < 0.01) and corneal nerve fiber length (*p* = 0.04) between T2DM-CKD and T2DM-no CKD groups. The two diabetes groups had reduced corneal nerve parameters compared to healthy controls (all parameters: *p* < 0.01). Immature central dendritic cell density was significantly higher in the T2DM-CKD group compared to the T2DM-no CKD group ((7.0 (3.8–12.8) and 3.5 (1.4–13.4) cells/mm^2^, respectively, *p* < 0.05). Likewise, central mature dendritic cell density was significantly higher in the T2DM-CKD group compared to the T2DM-no CKD group (0.8 (0.4–2.2) and 0.4 (0.6–1.1) cells/mm^2^, respectively, *p* = 0.02). Additionally, total central dendritic cell density was increased in the T2DM-CKD group compared to T2DM-no CKD group (10.4 (4.3–16.1) and 3.9 (2.1–21.0) cells/mm^2^, respectively, *p* = 0.03). Conclusion: The study showed that central corneal dendritic cell density is increased in T2DM-CKD compared to T2DM-no CKD, with groups matched for peripheral neuropathy severity. This is accompanied by a loss of central corneal nerve fibers. The findings raise the possibility of additional local factors exacerbating central corneal nerve injury in people with diabetic chronic kidney disease.

## 1. Introduction

The overlap of type 2 diabetes and chronic kidney disease (CKD) is considered a major risk factor for corneal nerve loss in people who have both conditions [1]. However, it is not known whether this greater corneal nerve loss in diabetic chronic kidney disease occurs in the setting of increased corneal inflammation as measured by corneal dendritic cell density.

Some studies have shown that diabetic corneal neuropathy is associated with increased corneal dendritic cell density [2,3], implying that corneal dendritic cells may have a role in corneal nerve regeneration or degeneration. In chronic kidney disease, uremic toxins with proinflammatory effects are released and their accumulation constitutes a persistent stimulus to systemic neuroinflammation [4,5]. Whether neuroinflammation is a significant contributor to corneal nerve damage in chronic kidney disease remains unclear.

The cornea is an excellent site to study neuroinflammation, as it receives the highest density of sensory nerves in the human nervous system [6]. In vivo corneal confocal microscopy of neuroimmune features that are reflective of corneal neuroinflammation permits visualization and quantification of both the corneal sub-basal nerve plexus and dendritic cells, their interactions and how these are affected in conditions such as chronic kidney disease [7]. Investigations undertaken in people with type 2 diabetes without chronic kidney disease have identified elevated corneal dendritic cell density compared to healthy controls [3]. However, it is not known whether compounding chronic kidney disease further alters corneal dendritic cell density, as is the case with corneal nerves [1]. Therefore, the aim of this study was to determine the impact of chronic kidney disease on corneal nerve measures and dendritic cell counts in type 2 diabetes.

## 2. Methods

The study received ethical approval from the South East Sydney Area Health Service and the University of New South Wales Research Ethics Committee. The procedures and protocols for the study were conducted according to the tenets of the Declaration of Helsinki (2013). Participants provided written informed consent following explanation of the protocols and procedures involved in the study. This was a cross-sectional study that included 29 participants with type 2 diabetes with chronic kidney disease and 29 participants with type 2 diabetes without chronic kidney disease. The two groups were first compared with 30 healthy controls to confirm that corneal nerve loss was greater in the presence of type 2 diabetes, as shown elsewhere [1]. Participants were recruited consecutively from the Diabetes Centre at the Prince of Wales Hospital, Sydney, Australia. Participants included in the study did not have current ocular infection, herpes simplex virus infection, corneal ectasia, corneal abrasion, history of refractive surgery, allergies to anesthetic eye drops, history of anterior segment trauma, and were not current contact lens wearers. Participants with a history of cataract surgery in the last six months were excluded. Participants were also excluded if they had a history of illnesses known to cause neuropathy and/or nephropathy, such as cancer, vitamin B12 deficiency, use of chemotherapy medications, previous neurotoxic therapy, treatment for neuropathic pain, carpal tunnel syndrome, peripheral edema, renal transplantation, glomerulonephritis, uncontrolled hypertension, or polycystic kidney disease. People were included only if they developed chronic kidney disease after the diagnosis of type 2 diabetes.

### 2.1. Study Participants

As there are no previous studies of corneal dendritic cell density in diabetic chronic kidney disease, sample size calculations were based on the corneal dendritic cell density in participants with diabetes (17.73 ± 1.45 cells/mm^2^) compared to controls (6.94 ± 1.58 cells/mm^2^) using G* Power 3.1.9.4 (Heinrich Heine University, Dusseldorf, Germany) [3]. A minimum sample of 20 participants in each group was adequate to detect a minimum mean difference of 1.7 ± 1.6 dendritic cells/mm^2^, with 80% power at an alpha level of 0.05.

### 2.2. Indicators of Metabolism and Kidney Function

In the current study, chronic kidney disease was defined as an estimated glomerular filtration rate less than 60 mL/min/1.73 m^2^, as well as mild to severe albuminuria based on Kidney Disease: Improving Global Outcomes guideline [8]. All participants with diabetes with chronic kidney disease had elevated serum creatinine and urea levels, as shown in Table 1 [8]. Participants with type 2 diabetes were categorized into two groups, namely, type 2 diabetes with chronic kidney disease (T2DM-CKD) and type 2 diabetes without chronic kidney disease (T2DM-no CKD). Serum urea, serum creatinine, serum potassium, albumin, serum triglycerides, total serum cholesterol, low-density lipoprotein, high-density lipoprotein, body mass index, HbA1c, liver function tests (aspartate transaminase, gamma-glutamyl transferase, alanine transaminase and alkaline phosphatase), bilirubin and total protein were collected at the time of examination from the electronic medical records.

### 2.3. Corneal Confocal Microscopy

Corneal staining was evaluated in both eyes with moistened fluorescein strips to ensure corneal integrity before corneal confocal microscopy. The right eye was imaged with the aid of a corneal confocal microscope (Heidelberg Retinal Tomograph III Rostock Cornea Module; Heidelberg Engineering GmbH, Heidelberg, Germany) [7,9]. The participants’ eyes were anaesthetized using a drop of 0.4% oxybuprocaine hydrochloride (Bausch & Lomb, Chatswood, Australia). A large drop of viscous liquid gel of 2.5% hydroxypropyl methylcellulose (GenTeal gel, Alcon Inc., Worth Fort, TX, USA) was applied onto the tip of the imaging lens and a clean Tomocap (Heidelberg Retinal Tomograph III Rostock Cornea Module; Heidelberg Engineering GmbH, Heidelberg, Germany) was inserted on the probe. The probe with the imaging lens was moved steadily until the Tomocap contacted the cornea.

Images were captured from the central cornea and inferior whorl region of the right eye only, as corneal nerve parameters have been shown to be symmetrical between eyes [10]. The inferior whorl region of the cornea is where the distal nerve endings of the cornea can be found [11,12]. This is where nerves converge 1–2 mm inferior to the corneal apex [11]. As corneal nerve loss in diabetes typically begins with the inferior whorl region before the central cornea [13], eight central and four inferior whorl images not overlapping by more than 20% for each participant were chosen for analysis [14]. Images were analyzed with the help of an automated nerve analysis software (Corneal Nerve Fiber Analyzer V.2, ACCMetrics, University of Manchester, Manchester, UK). The software estimates corneal nerve fiber density, corneal nerve fiber length, corneal nerve branch density, and inferior whorl length. Furthermore, the average nerve fiber length (ANFL) was calculated using the formula (corneal nerve fiber length + inferior whorl length/2), as per a previous study [15].

Corneal dendritic cells in the corneal sub-basal layer were defined as reflective bodies with dendritic processes [16]. Corneal dendritic cells were counted by two masked independent observers (KA and SA) using ImageJ Software (National Institutes of Health, Bethesda, MD, USA) [17]. The cells were categorized into immature dendritic cells (minute, reflective cell bodies with noticeable miniature dendrites with tapered ends (<25 µm)) and mature dendritic cells (reflective, slender cell bodies, often with dendritic processes extending out from the main cell body (>25 µm)) [15,16]. The average of eight images and four images in the central and inferior whorl cornea, respectively, were used in the analysis.

### 2.4. Peripheral Neuropathy Assessment

Previous studies have indicated that corneal nerve assessment may be abnormal in patients with peripheral neuropathy [18,19]. Therefore, the presence and severity of neuropathy was tested using the Total Neuropathy Score, a validated test for grading the severity of peripheral neuropathy in diabetes and chronic kidney disease [20,21]. The score assesses peripheral neuropathy across eight sections: motor and sensory neuropathic symptoms; pinprick sensibility; vibration detection; distal strength assessment; deep tendon reflexes and lower limb; sensory and motor nerve conduction studies [20]. Scoring for each domain ranged from 0 to 4, with 0 indicating no abnormality and 4 indicating greatest abnormality. Scores from each section were summed to give a Total Neuropathy Score between 0 and 32.

### 2.5. Data Analysis

Data were analyzed using SPSS version 23 (IBM Corp: Armonk, NY, USA) and Graph Pad Prism 9.0 (Graph Pad Software Inc., San Diego, CA, USA). Descriptive statistics were computed as means and standard deviations for continuous data, while categorical variables were computed as percentages and counts. Normal distribution of the data was assessed with the Shapiro–Wilk test, as well as the visual inspection of the quantile–quantile plots. Normally distributed data, such as the corneal nerve morphological parameters, were analyzed for three groups using the one-way fixed effects ANOVA with post hoc Bonferroni correction to determine any significant difference between the three group means. Due to differences in age between the three groups, pairwise comparisons were adjusted for age in the general linear model. For nonparametric data, such as dendritic cell parameters, a Mann–Whitney U test was used to determine any significant differences between the two diabetes groups. An independent samples *t* test or Mann–Whitney U was used to determine differences between other two-group comparisons based on normality test results. A *p* < 0.05 was considered statistically significant.

## 3. Results

### 3.1. Concordance between Masked Independent Observers

Figure 1 (Bland–Altman plots) illustrates the concordance between the masked independent observers (KA and SA) for dendritic cell density in the central cornea and the inferior whorl region, respectively. At both the central cornea and inferior whorl, agreement was consistent at the lower range and demonstrated a size effect with higher dendritic cell counts. This general observation was replicated at the two measurement locations. Data plots for the central cornea and inferior whorl have been presented to indicate the distribution of the data. Data presented below are the average of the two observers.

### 3.2. Participant Demographics, Metabolic Indicators and Measures of Ocular Surface Discomfort

Participant demographics, neuropathy severity and metabolic parameters are summarized in Table 1. Participants with T2DM-CKD and T2DM-no CKD were not significantly different in age, body mass index, duration of diabetes, HbA1c, serum triglycerides, total serum cholesterol, low-density lipoprotein, high-density lipoprotein and sex. T2DM-CKD and T2DM-no CKD groups were significantly different in serum potassium, serum creatinine, serum urea and urine albumin–creatinine ratio. There were no significant differences in the Total Neuropathy Score between T2DM-CKD and T2DM-no CKD, as the participants in each group were matched for Total Neuropathy Score (Table 1). There was a significant difference in age between T2DM-CKD, T2DM-no CKD and the healthy controls (70.6 ± 7.5, 66.4 ± 8.5 and 61.8 ± 5.9 years, respectively, F_(3,85)_ = 10.2, *p* < 0.01). However, post hoc testing with Bonferroni correction showed a significant difference only between T2DM-CKD and controls (*p* < 0.01). There was no significant difference in age between T2DM-CKD and T2DM-no CKD (*p* = 0.10), and T2DM-no CKD and controls (*p* = 0.06).

### 3.3. Corneal Nerve Parameters

One-way fixed effects ANOVA showed a significant difference between the three groups for corneal nerve fiber density (F_(3,85)_ = 31.3 *p* < 0.01), corneal nerve fiber length (F_(3,85)_ = 23.4 *p* < 0.01), corneal nerve branch density (F_(3,85)_ = 9.4 *p* < 0.01), inferior whorl length (F_(3,85)_ = 34.4 *p* < 0.01) and average corneal nerve length (F_(3,85)_ = 33.4 *p* < 0.01). Due to the differences in age between the three groups, the pairwise comparisons were adjusted for age in the general linear model.

There was a significant difference in corneal nerve fiber density (*p* < 0.01) and corneal nerve fiber length (*p* = 0.04) between T2DM-CKD and T2DM-no CKD. However, no significant differences were observed in average corneal nerve length (*p* = 0.35), corneal nerve branch density (*p* > 0.99), and inferior whorl length (*p* > 0.99) between T2DM-CKD and T2DM-no CKD (Table 2). In the T2DM-CKD group, there was a significant reduction in corneal nerve fiber density (*p* < 0.01), corneal nerve branch density (*p* = 0.03), corneal nerve fiber length (*p* = 0.01), average corneal nerve length (*p* < 0.01) and inferior whorl length (*p* < 0.01) compared to healthy controls. Similarly, in the group T2DM-no CKD, there was a significant reduction in corneal nerve fiber density (*p* < 0.01), corneal nerve branch density (*p* < 0.01), corneal nerve fiber length (*p* = 0.01), average corneal nerve fiber length (*p* < 0.01) and inferior whorl length (*p* < 0.01) compared to healthy controls.

### 3.4. Corneal Dendritic Cell Density

Immature central dendritic cell density was significantly higher in the T2DM-CKD group compared to the T2DM-no CKD group ((7.0 (3.8–12.8) and 3.5 (1.4–13.4) cells/mm^2^, respectively, *p* < 0.05). Likewise, central mature dendritic density was significantly higher in T2DM-CKD group compared to the T2DM-no CKD (0.8 (0.4–2.2) and 0.4 (0.6–1.1) cells/mm^2^, respectively, *p* = 0.02). Additionally, total central dendritic cell density was higher in the T2DM-CKD group compared to T2DM-no CKD group (10.4 (4.3–16.1) and 3.9 (2.1–21.0) cells/mm^2^, respectively, (*p* = 0.03), Table 3). This is shown in Figure 2.

In the inferior whorl region, mature dendritic cell density in the T2DM-CKD group in the inferior whorl was not significantly different to the T2DM-no CKD group (0.5 (0–1.5) and 0.0 (0.0–0.75) cells/mm^2^, respectively, (*p* = 0.07)). Likewise, the immature dendritic cell density was also not significantly different between T2DM-CKD and T2DM-no CKD groups (9.6 (2.8–22.8) and 10.3 (5.5 ± 13.8) cells/mm^2^, respectively, (*p* = 0.88), Table 2). This was also reflected in total dendritic cell density in the inferior whorl, which was not significantly different between T2DM-CKD and T2DM-no CKD groups (10.7 (2.8–21.8) and 10.3 (5.3–14.5) cells/mm^2^, respectively, (*p* = 0.72), Table 3).

There was a significant association between total central dendritic cell density and the following measures of kidney function: estimated glomerular filtration rate, (*rho* = −0.35, *p* < 0.01), serum creatinine (*r* = 0.4, *p* < 0.01) and urine–albumin–creatinine ratio (*r* = 0.35, *p* < 0.02). However, there were no associations between total central dendritic cell density and HbA1c (*r*ho = −0.06, *p* = 0.69.).

## 4. Discussion

The present study has assessed corneal nerve parameters and dendritic cell counts in patients with type 2 diabetes with and without chronic kidney disease. Methodologically, the study differs from previous work in this area, as both groups were matched for the severity of peripheral neuropathy, which has previously been considered to underlie the prominent changes in corneal nerve parameters [22]. This approach has resulted in the novel observation that the central cornea has more pronounced changes in terms of both nerve parameters and dendritic cell counts in the group with both diabetes and chronic kidney disease compared to the group with diabetes alone. The difference between the two groups does not appear to occur at the inferior whorl region of the cornea when groups are matched for neuropathy severity.

These observations also raise the question of whether there are additional factors within the central cornea that are driving nerve degeneration in the group with both diabetes and chronic kidney disease. The increased dendritic cell count in the central cornea noted in the group with chronic kidney disease raises the possibility that the presence of that condition is proinflammatory and may potentially be associated with the increased central corneal nerve fiber loss, independent of the presence or severity of peripheral neuropathy. A significant negative association between corneal nerve fiber length and branch density, and dendritic cell density in patients with type 1 diabetes, type 2 diabetes and latent autoimmune diabetes of adults has been reported, suggesting that inflammation may play a role in the development of corneal neuropathy [23]. Chronic kidney disease releases uremic toxins that instigate systemic neuroinflammation, which introduces further metabolic constraints on the central corneal nerves, leading to greater inflammation and central corneal nerve loss [24]. Alternatively, in a study by Pritchard and colleagues, a lower corneal nerve fiber length was found to be predictive of a four-year incidence of peripheral neuropathy in type 1 diabetes [25]. Based on those findings, a prospective study in diabetic chronic kidney disease may shed light on whether the increased central corneal nerve loss is predictive of the progression of generalized peripheral neuropathy.

The results of the current study showed that both immature and mature central corneal dendritic cells increase in diabetic chronic kidney disease compared to type 2 diabetes alone. Increased mature dendritic cells indicate more active inflammation, whilst immature cell increase indicates a more sentinel cell presence [3,15]. Overall, these findings indicate the presence of more active inflammation concurrent with greater corneal neuropathy in diabetic chronic kidney disease compared to type 2 diabetes alone.

The current study found differences in corneal nerve parameters and dendritic cells between groups in the central cornea but not in the region of the inferior whorl. Previous studies by our group and others have found the inferior whorl region to be more sensitive to neuropathic changes [13,26,27]. The possible explanation could be that a substantial reduction in the inferior whorl length may have already occurred in type 2 diabetes, even before the chronic kidney disease occurred. This substantial reduction has further decreased the variance between the groups. Furthermore, our group has also previously reported on the interobserver reliability of corneal nerve parameters in the central cornea versus the inferior whorl region and shown the latter to have poorer reliability, more so in corneas with more complex inferior whorl patterns [28]. The lack of nerve differences in this region between the two groups in the presence of central corneal differences may be due to this greater complexity in the inferior whorl region. With regards to dendritic cells, previous studies have reported a greater number of dendritic cells in the peripheral cornea, more so in the inferior and superior cornea [29,30], although these studies did not report on differences in the inferior whorl. The lack of difference found in the inferior whorl region between the two groups may reflect the dynamic and complex behavior of dendritic cells and other unknown factors [31].

From an ocular surface perspective, the loss of central corneal nerves and the presence of inflammatory dendritic cells in diabetic-related chronic kidney disease may contribute to increased susceptibility to neurotrophic ulcers [32], as well as dry eye disease [33]. The susceptibility to aqueous-deficient dry eye disease may be a result of a decreased stimulus for tear secretion by the lacrimal gland, as well as the increase in ocular surface inflammation [33]. People with chronic kidney disease undergoing hemodialysis have been found to have more dry-eye-related signs and symptoms compared to healthy controls [34,35,36,37], including lower tear volume, lower tear break-up time and tear meniscus height [34]. Proteins involved in inflammation and immune response have also been found to be elevated in the tear film of patients with chronic kidney disease [34]. Tear analysis in one study indicated that the differentially expressed proteins were involved in lipid metabolism, inflammation and immune responses [34], suggesting that changes in these pathways due to chronic kidney disease may be reflected in the ocular surface, contributing to the signs and symptoms of the ocular surface.

The current study has some limitations worth highlighting. The range of chronic kidney disease is wide, as are the ranges for corneal nerve parameters and dendritic cell counts. Hence, it would have been desirable for a subgroup analysis of chronic kidney disease stages to further strengthen the association between chronic kidney disease and dendritic cells counts. However, due to loss of matching for peripheral neuropathy severity and age, as well as loss of statistical power with chronic kidney disease severity grouping, this was not conducted in the current study. To remove any uncertainty that dendritic cell counts may not be discriminant between type 2 diabetes with and without chronic kidney disease, future studies with larger sample sizes can explore a paired analysis of corneal nerve loss and dendritic cells counts by study participant, as both groups showed portions of overlap in cornea nerve loss and dendritic cell counts. Moreover, future studies can explore the relationship between corneal neuroimmune features and other microvascular diseases in type 2 diabetes, such as diabetic retinopathy and diabetes-related microvascular cardiovascular disease.

In conclusion, the study showed that central corneal dendritic cell density is increased in diabetic chronic kidney disease compared to type 2 diabetes alone and that this is accompanied by a loss of central corneal nerve fibers. The findings raise the possibility of additional local factors exacerbating central corneal nerve injury in patients with diabetic chronic kidney disease.

## Figures and Tables

**Figure 1 jcm-12-00016-f001:**
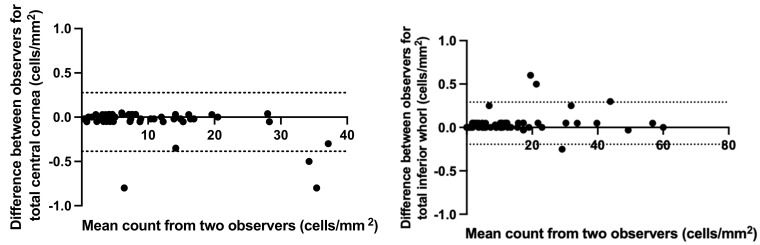
Agreement between total cell counts made by two observers from confocal microscopy images of the central cornea and inferior whorl. The mean difference was 0.05 ± 0.17 and 0.05 ± 0.12 in central and inferior whorl, respectively. Dotted lines show the upper and lower limits of agreement.

**Figure 2 jcm-12-00016-f002:**
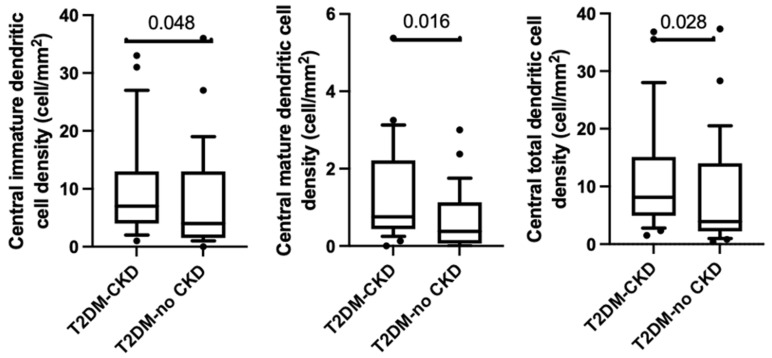
Boxplots and whiskers representing the 10th to 90th percentile of central immature dendritic cell density (ImDC), central mature dendritic cell density (MDC), and total central dendritic cell density (TotalDC).

**Table 1 jcm-12-00016-t001:** Clinical and metabolic indicators among patients with type 2 diabetes with (T2DM-CKD) and without (T2DM-No CKD) chronic kidney disease.

Parameter	T2DM-CKD	T2DM-No CKD	*p*-Value
Age, years	70.6 ± 7.5	66.4 ± 9.2	*p* = 0.06
Sex, % Male	65.5	79.3	*p* = 0.24
Body mass index, kg/m^2^	30.8 ± 7.4	31.8 ± 7.0	*p* = 0.61
Duration of diagnosis, years	19.3 ± 9.1	15.1 ± 12.0	*p* = 0.16
HbA1c, %	8.3 ± 1.9	8.8 ± 2.3	*p* = 0.43
Serum Urea, mg/dL	11.6 ± 4.8	6.0 ± 1.9	*p* < 0.01
Creatinine, mg/dL	200.3 ± 114.1	74.7 ± 138.2	*p* < 0.01
Estimated glomerular filtration rate, mL/min/1.73 m^2^	35.5 ± 17.8	82.0 ± 10.1	*p* < 0.01
Urine Albumin-creatinine ratio, mg/mmol	49.6 ± 92.1	3.8 ± 3.9	*p* < 0.01
Serum Potassium, mmol/L	4.6 ± 0.4	4.2 ± 0.5	*p* = 0.01
Alkaline phosphatase, U/L	78.3 ± 33.7	88.9 ± 26.7	*p* = 0.61
Gamma-glutamyl transferase, U/L	26.8 ± 14.5	58.0 ± 95.1	*p* = 0.20
Alanine transaminase, U/L	24.5 ± 17.1	32.6 ± 24.6	*p* = 0.24
Aspartate transaminase, U/L	26.5 ± 13.5	31.3 ± 26.3	*p* = 0.63
Albumin, g/L	38.1 ± 5.6	39.3 ± 5.6	*p* = 0.32
Total protein, g/L	70.6 ± 4.8	69.7± 6.1	*p* = 0.34
Bilirubin µmol/L	9.6 ± 5.7	10.2 ± 7.4	*p* = 0.62
Total cholesterol, mmol/l	3.8 ± 1.0	3.9 ± 1.2	*p* = 0.66
High density lipoprotein (HDL), mmol/L	1.1 ± 0.4	1.2 ± 0.4	*p* = 0.44
Low density Lipoprotein (LDL), mmol/L	1.9 ± 0.9	1.8 ± 1.1	*p* = 0.82
Triglycerides, mmol/L	2.2 ± 2.1	1.75 ± 1.4	*p* = 0.48
Total Neuropathy Score	8.9 ± 6.8	6.7 ± 5.4	*p* = 0.20

**Table 2 jcm-12-00016-t002:** Corneal nerve parameters among type 2 diabetes patients with and without chronic kidney disease.

Parameter	T2DM-CKD	T2DM-No CKD	*p*-Value
Corneal nerve fiber density (CNFD) (no./mm^2^)	12.5 ± 6.4	19.6 ± 7.1	*p* < 0.01
Corneal nerve branch density (CNBD) (no./mm^2^)	20.2 ± 20.6	19.6 ± 10.8	*p* > 0.99
Corneal nerve fiber length (CNFL) (mm/mm^2^)	9.0 ± 3.8	11.6 ± 3.6	*p* = 0.01
Inferior whorl length (IWL) (mm/mm^2^)	8.2 ± 3.6	8.9 ± 3.8	*p* = 0.54
Average corneal nerve fiber length (IWL + CNFL/2) (mm/mm^2^)	8.0 ± 3.8	9.7 ± 3.5	*p* = 0.09

**Table 3 jcm-12-00016-t003:** Dendritic cell density in type 2 diabetes with and without chronic kidney disease (median (interquartile range)).

Parameter	T2DM-CKD	T2DM-no CKD	*p*-Value
Central immature dendritic cell density (cells/mm^2^)	7.0 (3.8–12.8)	3.5 (1.4–13.4)	*p* < 0.05
Central mature dendritic cells density (cells/mm^2^)	0.8 (0.4–2.2)	0.4 (0.6–1.1)	*p* = 0.02
Total central dendritic cells density (cells/mm^2^)	10.4 (4.3–16.1)	3.9 (2.1–21.0)	*p* = 0.03
Inferior whorl immature dendritic cell density (cells/mm^2^)	9.6 (2.8–22.8)	10.3 (5.5 ± 13.8)	*p* = 0.88
Inferior mature dendritic cell (cells/mm^2^)	0.5 (0–1.5)	0.0 (0.0–0.75)	*p* = 0.07
Total whorl dendritic density (cells/mm^2^)	10.7 (2.8–21.8)	10.3 (5.3–14.5)	*p* = 0.72

## Data Availability

The data presented in this study are available on reasonable request from the corresponding author. The data are not publicly available due to ethical reasons.

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
