# Peer review of "Impact of Chronic Kidney Disease on Corneal Neuroimmune Features in Type 2 Diabetes"

_jcm, 2022, doi:10.3390/jcm12010016_

Round 1

Reviewer 1 Report

In this study, Asiedu et al. add to the relationship of diabetes and cornea taking the presence of CKD into account. The study is well presented and the association of CKD and not only NF loss but also increased dendritic cell density is of itself novel. However, some concerns are related especially to the study's power.

1. It is not clear, how many participants were finally included in the study and besides the eGFR cutoff of 60 ml, no other inclusion- or exclusion criteria are defined. 

2. Were all cases of CKD confirmed to be diabetic nephropathy? If so, by which methods was diabetic nephropathy confirmed?

3. The range of CKD is quite wide as are the ranges for NF and dendritic cell counts. Depending on the power of the study, subgroup analysis of CKD stages would be beneficial, to further strengthen the association of CKD and dendritic cells. Additionally, a paired analysis of NF and dendritic cells by study participant would be beneficial, too, as the diabetics with and without CKD show great portions of overlap in NF and dendritic cell counts rendering them statistically significant by their means, but non-discriminant.

Author Response

  1. In this study, Asiedu et al. add to the relationship of diabetes and cornea taking the presence of CKD into account. The study is well presented and the association of CKD and not only NF loss but also increased dendritic cell density is of itself novel. However, some concerns are related especially to the study's power.’

Authors’ response

Thank you for your comments. A sample size calculation was done to determine the minimum sample size required to demonstrate a difference in dendritic cell counts which is the primary outcome measure in the current study. Kindly see lines 86-90 “sample size calculations were based on the corneal dendritic cell density in participants with diabetes (17.73 ± 1.45 cells/mm2) compared to controls (6.94 ± 1.58 cells/mm2) using G* Power 3.1.9.4 (Heinrich Heine University, Dusseldorf, Germany).(3) A minimum sample of 20 participants in each group was adequate to detect a minimum mean difference of 1.7 ± 1.6 dendritic cells/mm2 with 80% power at an alpha level of 0.05.” Post-hoc power calculation based on the Mann-Whitney U test with 29 participants in each diabetic group could detect a minimum effect size of 0.54 at an alpha level 0.05 with an observed power of 0.8

  1. It is not clear, how many participants were finally included in the study and besides the eGFR cutoff of 60 ml, no other inclusion- or exclusion criteria are defined. 

Authors’ response

Thank you for your comment. There were 29 participants in the type 2 diabetes with chronic kidney disease group and 29 participants in the type 2 diabetes without chronic kidney disease group as well as 30 healthy controls. This is in line 66-70 “This was a cross-sectional study that included 29 participants with type 2 diabetes with chronic kidney disease and 29 participants with type 2 diabetes without chronic kidney disease. The two groups were first compared with 30 healthy controls to confirm that corneal nerve loss was greater in the presence of diabetes, as shown elsewhere.(1)”

The inclusion and exclusion criteria have been defined and this is on line 70-81 “People were included only if they developed chronic kidney disease after the diagnosis of type 2 diabetes. Participants were also excluded if they had a history of illnesses known to cause nephropathy such as renal transplantation, glomerulonephritis, uncontrolled hypertension, or polycystic kidney disease syndrome.”

  1. Were all cases of CKD confirmed to be diabetic nephropathy? If so, by which methods was diabetic nephropathy confirmed?

Authors’ response

Thank you for your comment. We only included participants who developed chronic kidney disease after the diagnosis of type 2 diabetes and who did not have any other known cause of chronic kidney disease as outlined on line 79-81 “Participants were also excluded if they had a history of illnesses known to cause nephropathy such as renal transplantation, glomerulonephritis, uncontrolled hypertension or polycystic kidney disease syndrome.”

  1. The range of CKD is quite wide as are the ranges for NF and dendritic cell counts. Depending on the power of the study, subgroup analysis of CKD stages would be beneficial, to further strengthen the association of CKD and dendritic cells. Additionally, a paired analysis of NF and dendritic cells by study participant would be beneficial, too, as the diabetics with and without CKD show great portions of overlap in NF and dendritic cell counts rendering them statistically significant by their means, but non-discriminant.

Authors’ response

Thank you very much for the comment. We agree with you on these comments however the aim of the current study was to determine the impact of chronic kidney disease on corneal nerve measures and dendritic cell counts as such we have matched for peripheral neuropathy severity and age between the two diabetes groups. We were not powered enough for a secondary analysis on the chronic kidney disease subgroups and the paired analysis. We have included a recommendation on line 320-324 in the paper on the subgroup and the paired analysis for future studies.

Reviewer 2 Report

Diabetic retinopathy is also an important indication of diabetes. Why you didn't describe the retinopathy of these patients. Is retinopathy related to corneal neuroimmune features?

Please keep all fonts consistent

Author Response

  1. Diabetic retinopathy is also an important indication of diabetes. Why didn't you describe the retinopathy of these patients. Is retinopathy related to corneal neuroimmune features?

Authors’ response

Thank you for your comment. Even though retinopathy or diabetic-related microvascular cardiovascular disease may or may not have a relationship with corneal neuro-immune features, it was not the focus of the current investigation. The aim of the current study was to explore the relationship between diabetic chronic kidney disease and corneal neuroimmune features. We have included these in our recommendations for future studies. Line 330-332 “Also, future studies can explore the relationship between corneal neuroimmune features and other microvascular diseases in type 2 diabetes such as diabetic retinopathy and diabetes-related microvascular cardiovascular disease.”

  1. Please keep all fonts consistent

Authors’ response

Thank you very much. All fonts have been formatted consistently.
